# Transmissive Mode Laser Micro-Ablation Performance of Ammonium Dinitramide-Based Liquid Propellant for Laser Micro-Thruster

**DOI:** 10.3390/mi14061219

**Published:** 2023-06-09

**Authors:** Baosheng Du, Yongzan Zheng, Chentao Mao, Haichao Cui, Jianhui Han, Luyun Jiang, Jifei Ye, Yanji Hong

**Affiliations:** State Key Laboratory of Laser Propulsion and Application, Department of Aerospace Science and Technology, Space Engineering University, Beijing 101416, China

**Keywords:** laser micro-ablation, ammonium dinitramide, near-infrared dye, ammonium perchlorate, energy deposition, energy conversion, laser micro-thruster

## Abstract

The transmissive mode laser micro-ablation performance of near-infrared (NIR) dye-optimized ammonium dinitramide (ADN)-based liquid propellant was investigated in laser plasma propulsion using a pulse YAG laser with 5 ns pulse width and 1064 nm wavelength. Miniature fiber optic near-infrared spectrometer, differential scanning calorimeter (DSC) and high-speed camera were used to study laser energy deposition, thermal analysis of ADN-based liquid propellants and the flow field evolution process, respectively. Experimental results indicate that two important factors, laser energy deposition efficiency and heat release from energetic liquid propellants, obviously affect the ablation performance. The results showed that the best ablation effect of 0.4 mL ADN solution dissolved in 0.6 mL dye solution (40%-AAD) liquid propellant was obtained with the ADN liquid propellant content increasing in the combustion chamber. Furthermore, adding 2% ammonium perchlorate (AP) solid powder gave rise to variations in the ablation volume and energetic properties of propellants, which enhanced the propellant enthalpy variable and burn rate. Based on the AP optimized laser ablation, the optimal single-pulse impulse (*I*)~9.8 μN·s, specific impulse (*I_sp_*)~234.9 s, impulse coupling coefficient (*C_m_*)~62.43 dyne/W and energy factor (*η*)~71.2% were obtained in 200 µm scale combustion chamber. This work would enable further improvements in the small volume and high integration of liquid propellant laser micro-thruster.

## 1. Introduction

In the past few decades, traditional hydrazine-based liquid propellants have dominated the space propulsion field [1]. However, in the long-term use process, many problems such as low specific impulse, flammability and explosion, environmental pollution, biological hazards, expensive launch equipment, and sustainable use have become increasingly prominent [2,3,4]. Due to the current global energy shortage and increasingly serious environmental problems, green and high-performance propellants have attracted more and more attention, and at the same time, higher requirements have been placed on the research and development of aerospace propellants [5]. Therefore, the development of pollution-free, recyclable, green and high-performance propellants has important scientific significance and application value. The ADN-based liquid propellant is a high-energy oxidant with a heat formation of 149.6 kJ/mol, it is halogen-free, has good chemical stability and safety, is pollution-free, and has a high performance, which has attracted more and more attention in the aerospace and civil fields, representing the research direction and development trend of new liquid propellants [6,7,8,9,10,11]. At present, the ADN-based liquid propellants used in space propulsion are mainly chemical propulsion, and the catalytic decomposition, ignition and combustion of ADN-based liquid propellants have been studied meticulously and extensively [12,13,14]. Nevertheless, chemical fuel propulsion has the disadvantages of requiring a catalyst, preheating, adjusting feed pressure, complicated combustion chamber structure and low specific impulse [2,15,16]. Traditional electric propulsion systems have drawbacks such as higher operating voltage and too low thrust [17]. In addition, due to the characteristics of small size, lightweight, low power consumption and low cost, micro-nano satellites pose new challenges for the propulsion system to be able to perform various tasks flexibly, such as long-term reliability, attitude control, orbit correction and orbit transfer [18,19,20]. Such propulsion systems must guarantee precision during operation, which can be achieved by generating tiny thrusts and impulses. As one of the advanced propulsion technologies, laser ablation propulsion technology refers to the use of the interaction between the laser and the material from a long distance to generate high-temperature and high-pressure gas, which is sprayed through the nozzle to generate thrust. Compared with chemical fuel propulsion and electric propulsion, laser pulsed propulsion technology with improved propulsion efficiency, reduced launch cost and increased payload has become a promising propulsion technology due to its many inherent advantages [21,22,23,24]. For instance, Myrabo et al. successfully propelled a light vehicle with a mass of 50 g and a diameter of 12.2 cm to a height of 71 m using a 10 kw laser pulse [25]. The results show that the propulsion efficiency depends on the laser energy and the mass of the light vehicle. With the further development of laser plasma propulsion technology, a large amount of research work has gradually shifted from the macro field to the micro field [26,27,28,29,30]. 

So far, studies on the interaction of pulsed lasers with liquid propellants have been reported in a large number of studies in the literature [31,32,33]. The interaction between the laser pulse-induced plasma (or shock wave) and the material is the physical mechanism for the performance of pulsed laser ablation [21,22,23,24,34]. To further improve the ablation performance of pulsed lasers for liquid propellants, efforts have been made to synergistically optimize laser ablation performance by coupling laser energy into energetic propellants [35,36,37,38,39]. However, considering the solubility and oxidation resistance of laser absorbers, laser energy deposition of energetic liquid propellants at the micrometer scale is still difficult to achieve. In addition, laser incidence mode and the micro-scale combustion chamber configuration also have a crucial impact on the laser micro-ablation performance. There are still a lot of key technologies to be solved for the laser micro-thruster, such as the space-borne high-power laser technology, the development of high-performance propellants, the design of laser optical path and the integrated design of space micro-thruster, etc. [40].

In this work, the laser micro-ablation performance of near-infrared dye-optimized ADN-based liquid propellants was investigated using 200 µm combustion chambers. The results show that micro-ablation performance depends on the synergistic effect of laser energy deposition efficiency and chemical energy release from ADN-based liquid propellants, exhibiting high specific impulse and energy factors. The physical mechanism of the impulse generated by the interaction between the laser and the ADN-based liquid propellant is investigated. Potential applications of laser micro-ablation properties in the field of micro-thruster are discussed.

## 2. Materials and Methods

### 2.1. Experimental Items and Devices

ADN-based liquid propellant solution (63.4% and solid powder, 11.2% methanol and 25.4% water) was provided by the Dalian Institute of Chemical Physics. The acetone and other reagents required for the experiment were purchased from J&K Scientific Ltd. Analytical grade (99%) ammonium perchlorate was obtained from Shanghai Aladdin Biochemical Technology Co., Ltd. (Shanghai, China). The e2057 type near-infrared absorbing dye was purchased from Epolin INC. The combustion chamber array chip made of pure quartz materials (rectangular-200 × 200 µm^2^,) was processed by Suzhou Chuangkuo Metal Technology Co., Ltd. (Suzhou, China). The three-axis stage (OMSC40-20, Stroke 21 mm) was employed to adjust the spatial position of the solid-state YAG laser, and the repeat step accuracy of X, Y and Z axes are all <2 µm. The displacement sensor (Wenglor PNBC001, working range 20~24 mm, resolution 0.06 μm, linearity deviation 2 μm, response time < 33 μs, output frequency 10~30 kHz) was used to measure the displacement of the torsional pendulum from the equilibrium position. A digital microscope (DinoLite AM7515, resolution 2592 × 1944, frame rate 30 fps, magnification 20–220×) was utilized for laser focusing and observation of flow field evolution. A high-speed camera (Phantom V711, maximum pixel 1280 × 800, 7530 fps, maximum shooting speed 1.4 Mfps, corresponding pixel 128 × 8, minimum exposure time 1 μs) was applied to laser focusing and observing the evolution of the flow field. The laser-related parameters were measured by the oscilloscope (Tektronix mdo3054, bandwidth 500 MHz, sampling rate 2.5 gs/s). The signal generator (DG645, resolution 5 ps, accuracy 1 ns, jitter external trigger < 25 ps, internal trigger < 15 ps, external trigger delay 85 ns) was used for an external trigger of laser and high-speed camera. The delay time between the trigger signal and plasma generation was detected by a high-speed photodiode (Thorlabs, wavelength range 200~1100 nm, rising edge < 300 ps, falling edge < 500 ps, cut-off frequency > 1 GHz). Solid-state YAG lasers and semiconductor lasers are customized by the China Electronics Technology Group Corporation with a central wavelength of 1064 nm, a peak power density of ~1 × 10^10^ W/cm^2^, a single pulse energy of 20 mJ, a focus spot size ~258 μm, a focal length of 13.5 mm, and a pulse width of 5 ns.

### 2.2. Liquid Propellant Formulation

ADN-based liquid propellant formulations are 0%-AAD (15 mg dye powder in 1 mL acetone), 20%-AAD (0.2 mL ADN solution + 0.8 mL dye solution), 30%-AAD (0.3 mL ADN solution + 0.7 mL dye solution), 40%-AAD (0.4 mL ADN solution + 0.6 mL dye solution), 50%-AAD (0.5 mL ADN solution + 0.5 mL dye solution), 60%-AAD (0.6 mL ADN solution + 0.4 mL dye solution), 80%-AAD (0.8 mL ADN solution + 0.2 mL dye solution), 100%-AAD (1 mL ADN solution). The liquid propellant formulation for the propulsion performance contained three parts: ADN-based liquid propellant, ammonium perchlorate and near-infrared dye (the concentration of the dye). 

### 2.3. Absorption Depth Measurement of Liquid Propellants 

A near-infrared spectrometer was employed to measure the transmittance of different ratios of ADN-based liquid propellant solutions from 900 to 1700 nm bandwidth. Firstly, the prepared ADN-based liquid propellant solution was pipetted into a 0.1 mm optical path cuvette. Secondly, the cuvette was placed into the spectrometer, and then detected the transmittance of the propellant in the air atmosphere as the reference data of the testing background. Then, the transmittance of different ratios of ADN-based liquid propellant solutions was tested in turn. Finally, the transmittance at 1064 nm was recorded and the absorption coefficient and absorption depth of the samples were calculated according to the Beer–Lambert law. The Beer–Lambert law formula is as follows: (1)$$I_{transmittance}=I_{0}e^{-rx}\;$$

Among them, *I_transmittance_* is the energy of the laser after passing through the liquid working medium, *I*_0_ is the outgoing energy of the laser, *r* is the absorption coefficient, and *x* is the optical path. The absorption depth (1/*r*) is the inverse of the absorption coefficient.

### 2.4. Thermal Analysis Measurement of Liquid Propellants 

A differential scanning calorimeter was applied to test the enthalpy variable of ADN-based liquid propellant solutions with different ratios. The enthalpy variable refers to the energy difference between the product and the reactant. The calculation method is as follows: (2)$$\Delta H=E_{absorption}-E_{release}=H_{reaction\;product}-H_{reactant}$$
where *E_absorption_* represents the total energy absorbed when the reactant breaks bonds and *E_release_* represents the total energy released when the product forms a bond. when Δ*H* is “+”, it means an endothermic reaction, and when Δ*H* is “−”, it means an exothermic reaction. An amount of 15 mg liquid was weighed for each sample and configured samples were placed in an alumina ceramic crucible, respectively. The test temperature was 60–300 °C, the heating rate was 10 °C/min, and the test condition was carried out in nitrogen atmosphere conditions. Then, 20%-AAD, 30%-AAD, 40%-AAD, 50%-AAD, 60%-AAD, and 80%-AAD liquid propellant solutions were utilized to measure the curve of differential scanning calorimeter (DSC) as a function of temperature for thermal analysis.

### 2.5. Ablation Characteristics Measurement with Torsion Pendulum

A torsional thrust stand composed of torsional arm, flexural pivots and electromagnetic damping is developed by our team for measuring ablation characteristics [41]. When an impulsive force *I* is applied to the torsional thrust stand, the maximum angular displacement $\theta_{\max}$ can be given by:(3)$$\theta_{\max}=\frac{I\;L_{f}}{Jw_{d}}e^{-\frac{2\zeta }{\pi \sqrt{1-\zeta^{2}}}}$$
where J, $w_{d}$ and ζ are the moment of inertia, the damping ratio and the natural frequency of the torsional thrust stand, respectively. These system parameters can be obtained by the calibration method called the free vibration method. $L_{f}$ is the distance between the rotational axis and the location of impulsive force.

Re-arranging Equation (3) with the consideration of small angle approximation, the impulsive force can be obtained by:(4)$$I=\frac{\theta_{\max}Jw_{d}}{L_{f}}e^{\frac{\pi \zeta }{2\sqrt{1-\zeta^{2}}}}$$

The maximum angle displacement can be given by the maximum linear displacement $x_{\max}$, i.e.,
(5)$$\theta_{\max}=x_{\max}/L_{m}$$
where $L_{m}$ is the distance between the rotational axis and the linear displacement sensor.

Then
(6)$$I=\frac{x_{\max}Jw_{d}}{L_{m}L_{f}}e^{\frac{\pi \zeta }{2\sqrt{1-\zeta^{2}}}}$$

The specific impulse (*I_sp_*) is further calculated on the basis of the impulse *I*, assuming that the liquid propellant in the micro-ablation cavity is completely ablated, and the mass of ablation (*m*) = the volume of the ablation cavity × the density of the liquid propellant. The specific impulse is calculated according to the following formula: (7)$$I_{sp}=I/mg=v/g$$

The impulse coupling coefficient (*C*_*m*_) is calculated by combining the impulse *I* and the laser output energy (*E*), and the impulse coupling coefficient is calculated according to the following formula: (8)$$C_{m}=\frac{I}{E}=mv/E$$

The energy conversion efficiency can be calculated by the specific impulse and the impulse coupling coefficient. The calculated energy system is represented by *η*, and the energy factor is calculated according to the following formula: (9)$$\eta =\frac{\frac{1}{2}m\nu^{2}}{E_{\mathrm{laser}}}=\frac{g}{2}C_{m}I_{sp}$$

In this work, the torsional pendulum measurement method was employed to measure the single-pulse impulse of laser ablation of ADN-based liquid propellant, and the experimental system was set up as shown in Figure 1. The solid-state YAG laser with a pulse width of 10 ns and energy of 20 mJ is applied. The laser displacement sensor model was purchased from Wenglor (PNBC001), with 30 kHz frequency, 4 mm maximum range and 0.0003% resolution. The laser is fixed on a three-axis stage. The position of the laser can be adjusted by controlling the stage with 1 μm movement accuracy. A tiny combustor chip is placed on each end of the torsion, then the liquid propellant filled in the micro combustor is ablated by the laser. The impulse generated by ablating the liquid propellant leads the torsional pendulum oscillating, and the position changes are recorded by the displacement sensor, and then calculated by the torsional pendulum measurement system. A photodetector is placed on the side of the laser light outlet to capture the light signal and transmit the signal to DG645. After receiving the signal, DG645 sends a signal to the high-speed camera to capture and photograph the evolution of the plume. Oscilloscope is utilized to record electrical signals from signal generators and photodetectors.

## 3. Results and Discussions

### 3.1. Laser Energy Deposition Characteristics of ADN-Based Liquid Propellants

A near-infrared spectrometer was used to measure the transmittance of different ratios of ADN-based liquid propellant solutions (0%-AAD, 20%-AAD, 30%-AAD, 40%-AAD, 50%-AAD, 60%-AAD, 80%-AAD, and 100%-AAD). The individual sample test spectra are organized in Figure 2a. In this paper, the micro-ablation performance of ADN-based liquid propulsion is mainly studied by solid-state YAG laser. Therefore, the laser energy deposition efficiency of liquid propellants at 1064 nm is extremely important. Unlike traditional laser propulsion, a large volume of liquid propellant can provide sufficient depth to deposit the energy of the laser, while in laser micro-propulsion research, the spatial depth of the liquid propellant is extremely small and cannot effectively transfer the laser energy deposited into the liquid propellant. For this reason, near-infrared dyes with high absorption ability can effectively optimize the problem of laser energy deposition during laser micro-ablation. As can be seen in Figure 2a, the volume fraction of the near-infrared dye in acetone solution increases, the energy deposition effect of ADN-based liquid propellant is also significantly enhanced, and the light absorption frequency band gradually widens. The transmittance, absorption coefficient and absorption depth obtained for each sample at 1064 nm were recorded in Figure 2b–d. As shown in Figure 2b, the transmittance of 0%-AAD, 20%-AAD, 30%-AAD, 40%-AAD, 50%-AAD, 60%-AAD, 80%-AAD, and 100%-AAD are reached 0.2%, 5.4%, 9.7%, 13.4%, 31.3%, 48.7%, 74.1%, and 93.0%, respectively. The experimental data show that the energy of the YAG laser can be deposited by appropriately adjusting the volume fraction of near-infrared dye in acetone solution within a 100 μm optical path. According to Beer–Lambert law, the absorption coefficient decreases with increasing ADN solution volume fraction, while the absorption depth increases with increasing ADN solution volume fraction (Figure 2c,d).

### 3.2. Energetic and Thermal Release Properties of ADN-Based Liquid Propellants

A differential scanning calorimeter was used to measure the DSC curves as a function of temperature with different proportions of ADN-based liquid propellants (0%-AAD, 20%-AAD, 30%-AAD, 40%-AAD, 50%-AAD, 60%-AAD, 80%-AAD, and 100%-AAD). The experimental results are presented in Figure 3a–f. The exothermic peak temperature and enthalpy variables of each sample were obtained through testing and sorted out in Table 1 for analyzing the energetic and thermal release properties of ADN-based liquid propellants.

The enthalpy variable of ADN-based liquid propellants increases with increasing ADN content. The enthalpy variables of 0%-AAD, 20%-AAD, 30%-AAD, 40%-AAD, 50%-AAD, 60%-AAD, 80%-AAD, and 100%-AAD were +108.8 J/g, −303.3 J/g, −313.2 J/g, −345.7 J/g, −368.3 J/g, −371.7 J/g, −425.4 J/g, −432.2 J/g, respectively. The dye solution without ADN liquid propellant only has an obvious endothermic peak around 100 °C, and there is no obvious exothermic peak between 60–300 °C. However, the enthalpy variable value of the ADN liquid propellant without the dye solution is the highest. The exothermic peaks of ADN-based liquid propellants are all around 190 °C, and the exothermic peaks are relatively sharp, which is conducive to the rapid heat release of energetic ADN-based liquid propellants in a short time to form a high specific impulse. The liquid propellant with less ADN content and more dye solution has two obvious square heat peaks, and the temperature of the exothermic peak is higher. The experimental results show that the exothermic peak temperature of the liquid propellant with high acetone solution content of dye is higher. Too much dye can affect the exothermic temperature of the liquid propellant.

### 3.3. Laser Micro-Ablation Performance of ADN-Based Liquid Propellants

In this work, the material of the micro-ablation cavity is pure quartz glass and micro-combustor arrays were machined on both ends of the chip (Figure 4a–c). Firstly, the micro-combustors cavity array was fabricated on one end of a 20 × 20 × 2 mm^3^ quartz glass sheet, then another quartz glass sheet was combined by plasma bonding. As shown in Figure 4b, the direction in which the incident laser light is defined as the *x*-axis direction, and the direction in which the laser ablation products are being sprayed is defined as the *y*-axis direction. The ablation combustor is about 200 × 200 µm^2^ square when viewed from the *x*-axis direction (Figure 4b), and the parabolic combustor with a depth of about 200 µm when viewed from the *y*-axis direction. The ablation chambers were configured as parabolic cylinders, and the volume of a single combustor was approximately 5.33 nL (Figure 4c).

In order to evaluate the ablation performance of ADN-based liquid propellants with different ratios, the torsional pendulum method was applied to measure the single-pulse impulse of the laser ablation interaction with ADN-based liquid propellant using transmission mode. According to Equation (6), if the maximum displacement (*x*_max_) of the combustion chamber chip from the equilibrium position is measured by the displacement sensor, the single-pulse impulse of the liquid propellant can be obtained. Figure 5a–f shows the displacement versus time after the interaction of the laser with the ADN-based liquid propellant using the YAG laser in a parabolic combustion chamber. As can be seen from Figure 5, as the ADN solution content increased from 20%-AAD to 80%-AAD, the displacement first increased and then decreased. The displacement of the 40%-AAD liquid propellant reached the maximum value, which was attributed to the single-pulse impulse of the liquid propellant, is affected by two main factors: the laser energy deposition efficiency and the energy content of the liquid propellant. However, no obvious displacement was detected for the liquid propellants of 0%-AAD and 100%-AAD, which further indicates that during the interaction between the laser and the liquid propellant, simply increasing the laser deposition efficiency and the energy of the liquid propellant cannot obtain excellent single-pulse impulse. 

There is no displacement was detected in the 0%-AAD and 100%-AAD liquid propellants. Thus, no impulse is generated. The 100%-AAD liquid propellant has basically zero absorption rate for the 1064 nm laser, which leads to the absence of laser energy deposition and then no single-pulse impulse. Figure 6a shows that the proportion of ADN has a great influence on the single-pulse impulse, and it increases first and then decreases with the increase in the proportion of ADN. The liquid propellant impulse of 40%-AAD achieved the maximum thrust (~6.4 μN·s). The main reason is that the absorption rate of laser light and the chemical energy contained in liquid propellants are different. 

The specific impulse (*I*_*sp*_) is further calculated on the basis of the single-pulse impulse. Assuming that all the liquid propellant in the micro-ablation cavity is ablated, the mass of ablation = the volume of the ablation cavity × the density of the liquid propellant. The obtained specific impulse results are shown in Figure 6b. The specific impulse of the 40%-AAD liquid propellant is the largest, reaching ~153.7 s, and then continuously decreasing to about 54.2 s (80%-AAD). Combining the impulse and laser parameters, the impulse coupling coefficient (*C*_*m*_) can be calculated, the average value of the laser energy is ~20 mJ. As shown in Figure 6c, the obtained 40%-AAD liquid propellant has the largest *C*_*m*_, increasing to 31.6 dyne/W, and then gradually decreasing to 11.1 dyne/W. The energy factor (*η*) can be calculated by combining the *I*_*sp*_ and *C*_*m*_. Figure 6d present *η* of ADN-based liquid propellants with different ratios. The obtained 40%-AAD liquid propellant has the largest *η*, increasing to 24.4%, and then gradually decreasing to 3.1%.

In order to further improve the laser micro-ablation performance of ADN-based liquid propellants, the ablation performance of the liquid propellants with and without added of 2 wt.% ammonium perchlorate (AP) solid powder was compared under the same experimental conditions. Figure 7a shows the time-dependent displacement of the YAG laser interacting with the liquid propellants of 40%-AAD and 40%-AAD + AP. The experimental results show that the maximum displacement of the liquid propellant with AP is significantly larger than that of the liquid propellant without AP. Finally, the maximum displacement obtained by multiple measurements under each condition is substituted into Equation (6) to calculate the single-pulse impulse.

As shown in Figure 7b, the *I*_*sp*_ of 40%-AAD liquid propellant achieved ~153.7 s. The *I*_*sp*_ of 40%-AAD + AP liquid propellant reaches ~234.9 s when AP was added. Similarly, the *C*_*m*_ of 40%-AAD + AP liquid propellant was greatly improved to reach ~62.4 dyne/W, which was much higher than the *C*_*m*_ of 40%-AAD liquid propellant reached ~31.6 dyne/W (Figure 7c). The addition of AP also enhanced the *η* of ADN-based liquid propellants. As shown in Figure 7d, the *η* of 40%-AAD liquid propellant achieved ~24.4%, then increased to ~71.2% after the AP was added. The reason for using AP is its high burning rate, non-hygroscopic crystals, excellent burning rate regulation ability, and easy compatibility with various fuels. In addition, the decomposition reaction of AP can generate oxygen to provide the necessary conditions for the rapid release of chemical energy of ADN-based liquid propellant [41,42,43]. The reaction equation is as follows:(10)$${2\mathrm{NH}}_{4}{\mathrm{ClO}}_{4}{=N}_{2}↑{+4H}_{2}{O+\mathrm{Cl}}_{2}↑{+2O}_{2}↑$$

For comparison, the laser micro-ablation characteristics of different liquid propellants under this working condition, the single-pulse impulse test data of water, methanol, acetone, acetone + dye, 40%-AAD liquid propellant and 40%-AAD + AP liquid propellant are shown in Figure 8. The single-pulse impulses of water, methanol, and acetone are close to zero. The results show that under this working condition, liquids such as water, methanol, and acetone in the ADN-based liquid propellant do not contribute to the impulse generated by the interaction between the laser and the liquid. The main functions of these three liquids in propellants are to dissolve energetic items and dyes and assist combustion to form plasma. 

### 3.4. Mechanism Analysis of Laser Micro-Ablation of ADN-Based Liquid Propellants

To clarify the mechanism of the impulse generated by the interaction between single-pulse laser and ADN-based liquid propellants, we first studied the effect of ADN-based liquid propellants optimized by different content of near-infrared dye solutions on the laser energy deposition efficiency. Secondly, the energy content and heat release characteristics of ADN-based liquid propellants were investigated by thermal analysis. Based on the analysis of the experimentally measured impulse results, the impulse generated by laser ablation of ADN-based liquid propellants at the 200-micron scale is jointly determined by the laser energy deposition efficiency and the propellant’s energetic properties. Unilaterally increasing the laser energy deposition efficiency of the propellant and the energetic properties of the propellant are not conducive to generating impulse. To further study the performance of laser micro-ablation, the schlieren test method was used to analyze the evolution of the flow field after the interaction between the laser and the liquid propellant. Figure 9 is a schlieren image of the 40%-AAD and 40%-AAD + AP propellant samples. From the results, an obvious shock wave begins to be generated at 1 μs. The 40%-AAD propellant sample is not fully ionized under the action of the laser. The jet plume is ejected from the nozzle, the semicircular shock wave before the plume is ejected, and the shock wave formed by the laser is rapidly ejected outward in the direction of the nozzle. Comparing the schlieren images of the 40%-AAD + AP propellant samples, a stronger shock wave was generated at the same time, and the shock wave formed an ellipse in the direction of the nozzle behind the jet plume. The above experimental results further show that the AP-added liquid propellant has a faster plume ejection velocity, which in turn generates a larger impulse [42,43].

As shown in Figure 10, the mass of the 40%-AAD + AP propellant sample decreases gradually with the increase in temperature. The mass loss between room temperature and 180 °C is mainly due to the volatilization of solvent in the propellant. The mass of the propellant decreases rapidly around 192 °C, which is due to the rapid exothermic reaction of energetic groups in the propellant. In the range of 230–500 °C, the mass of propellant decreases slowly with the further increase in temperature, and the mass of propellant products is only ~10%. This shows that the rapid exothermic reaction consumes most of the propellant mass, and the propellant ablation products are less. The above experimental results indicate that ADN-based liquid propulsion can quickly release chemical energy, thereby generating high temperature and pressure in the microscale combustion chamber, which is conducive to the formation of greater impulses.

Figure 11 shows the photoelectric signal in the process of single pulse laser ablation of and-based liquid propellant by a photodiode. It can be seen from the figure that the width of the laser diode pump light is 230 µs, and then the signals of laser and plasma are detected (230–1500 µs), which indicates that the high-power density laser ablation of micron-scale liquid propellant produces plasma, thus reducing the liquid splash, which is helpful to improve the laser micro-ablation performance (as shown in Appendix A).

Therefore, based on the above experimental results, the excellent laser ablation performance of the propellant sample of 40%-AAD + AP in a 200 µm scale combustion chamber includes: (1) the laser energy deposition efficiency of the propellant is much greater (as shown in Figure 1); (2) the propellant contains more energy and can rapidly release heat under low-temperature conditions (as shown in Figure 2 and Figure 10); (3) adding AP can increase the energy content and heat release rate of the propellant, thereby forming high temperature and high pressure in the combustion chamber to generate a greater impulse. 

## 4. Conclusions and Future Work

In this work, near-infrared dye-optimized ADN-based liquid propellants with different volume content were performed to investigate the effects of transmissive mode laser micro-ablation performance using a 200 µm combustion chamber. As the volume ratios of ADN liquid propellant increased, the transmissive mode laser micro-ablation performance first increased and then gradually decreased, and the best performance of 40%-AAD liquid propellant was obtained. The laser micro-ablation performance of different liquid propellants such as water, methanol, acetone, acetone solution of dyes, and ADN liquid propellant was compared. It is proved that the laser energy deposition coefficient and the propellant chemical energy release are the two most important factors for laser micro-ablation performance. The effect of the addition of AP on the performance of laser micro-ablation was studied under the optimal ratio of 40%-AAD liquid propellant. The experimental results show that the ablation performance parameters of the liquid propellant-added AP are obviously better than those without AP. 

The experimental results presented here can be expected to have practical applications in the design of laser micro-thrusters with ADN-based liquid propellants. Especially in comparison with the ablation performance of other liquid propellants (Table 2), the laser micro-ablation performance of ADN-based liquid propellant is more outstanding while more economical propellant. Although their composites show better performance in ablation performance, our method presents certain advantages in combustion chamber configuration and laser path design. The transmission mode laser ablation and micron-scale combustion chamber are more contributed to the design of small volume and high specific impulse laser micro-thruster. Future work concerning ADN-based liquid propellants will focus on the 1U laser micro-thruster development. Transmission mode laser micro ablation of liquid propellant provides an effective scheme for the integrated technology of laser micro-thruster because of its simple structure and not easy to be contaminated by ablation jet. However, the combustion chamber structure design, the laser window material and high-performance liquid working medium development of laser micro-thruster are still huge challenges. We look forward to the practical applications of pulsed laser micro-thruster in aerospace projects, deep space exploration, space missions, and other exciting fields.

## Figures and Tables

**Figure 1 micromachines-14-01219-f001:**
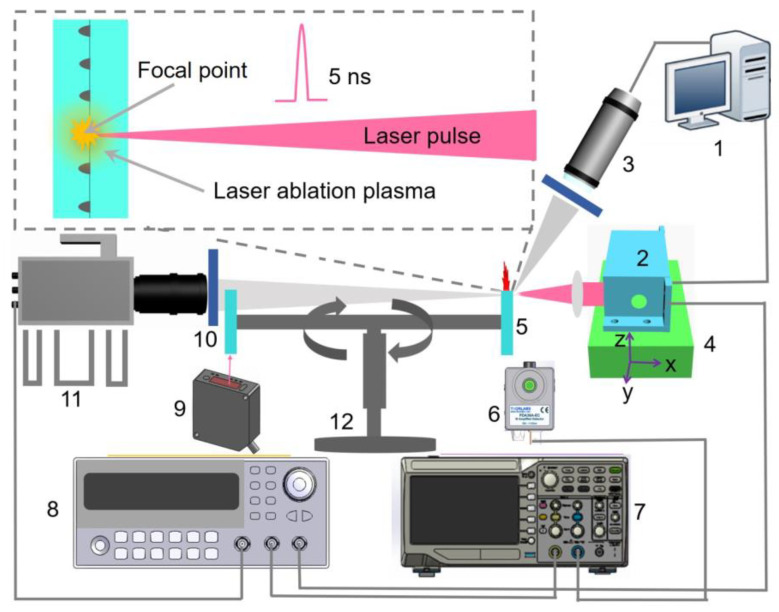
Single pulse impulse and flow field evolution process measurement system. (1) Remote control system; (2) laser; (3) digital microscope; (4) three-axis translation stage; (5) combustion chamber chip fixtures; (6) photodetector; (7) oscilloscope; (8) signal generator; (9) displacement sensor; (10) filter; (11) high-speed camera; (12) torsional thrust stand.

**Figure 2 micromachines-14-01219-f002:**
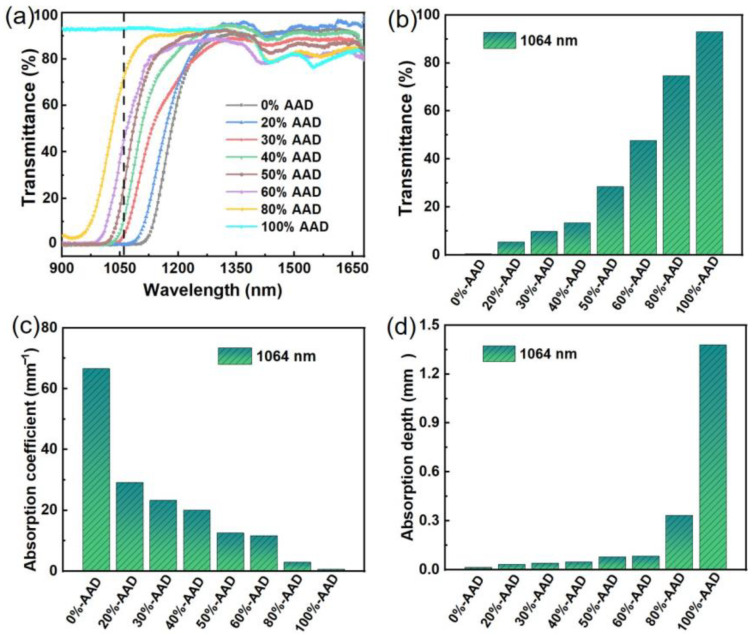
Transmittance curves with wavelength (**a**), transmittance (**b**), absorption coefficient (**c**), and absorption depth (**d**) at 1064 nm of propellant samples of 0%-AAD, 20%-AAD, 30%-AAD, 40%-AAD, 50%-AAD, 60%-AAD, 80%-AAD, and 100%-AAD.

**Figure 3 micromachines-14-01219-f003:**
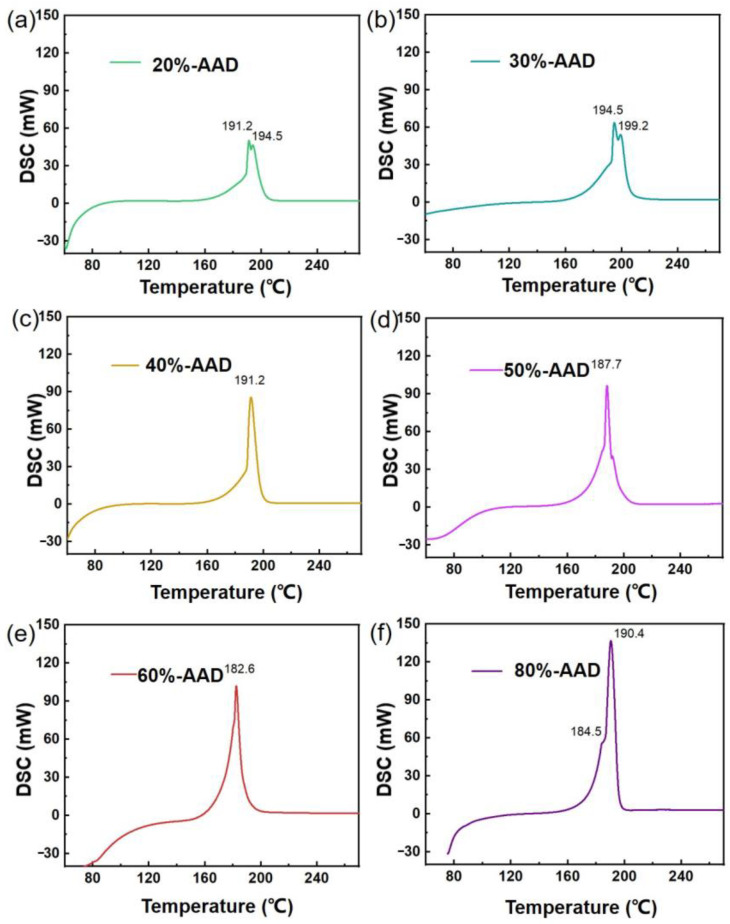
DSC curves of propellant samples of 20%-AAD (**a**); 30%-AAD (**b**); 40%-AAD (**c**); 50%-AAD (**d**); 60%-AAD (**e**); and 80%-AAD (**f**) with same heating rates.

**Figure 4 micromachines-14-01219-f004:**
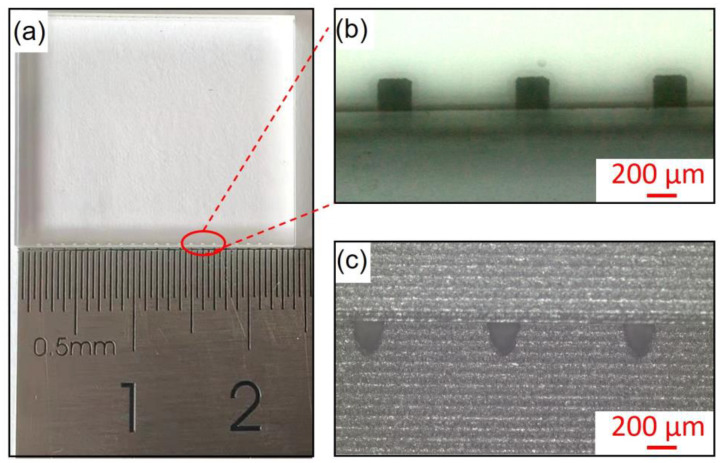
Structure diagram of micro combustor chip (**a**), top view (*x*-axis direction) of the combustion chamber cavity (**b**), and profile view (*y*-axis direction) of the parabolic combustion chamber cavity (**c**).

**Figure 5 micromachines-14-01219-f005:**
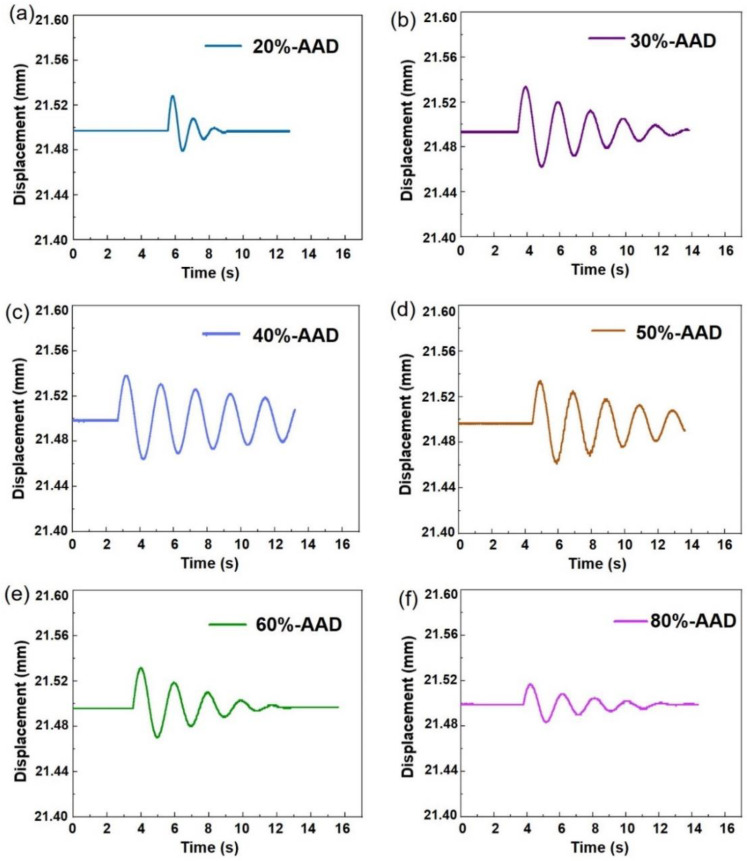
Displacement curves of the propellant samples over time 20%-AAD (**a**), 30%-AAD (**b**), 40%-AAD (**c**), 50%-AAD (**d**), 60%-AAD (**e**), and 80%-AAD (**f**).

**Figure 6 micromachines-14-01219-f006:**
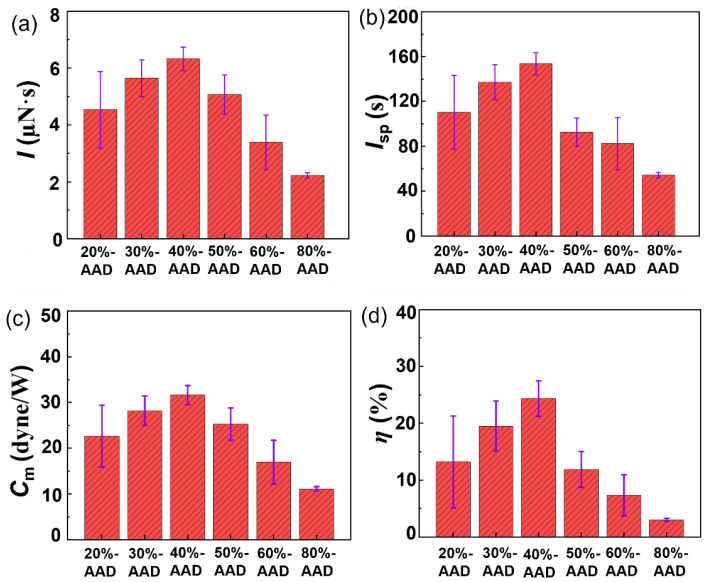
Experiment results of single-pulse impulse (**a**), specific impulse (**b**), impulse coupling coefficients (**c**), and energy factor (**d**) of different propellant samples.

**Figure 7 micromachines-14-01219-f007:**
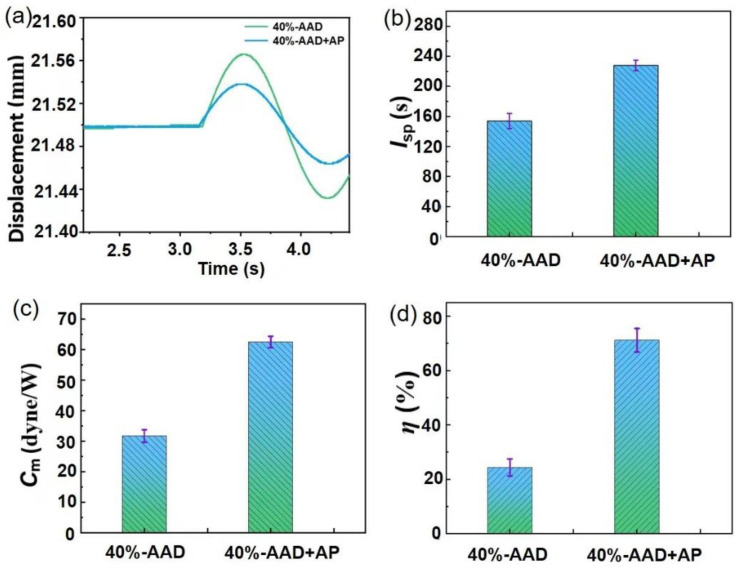
Experiment results of displacement curves (**a**), specific impulse (**b**), impulse coupling coefficients (**c**) and energy factor (**d**) of 40%-AAD without and with AP.

**Figure 8 micromachines-14-01219-f008:**
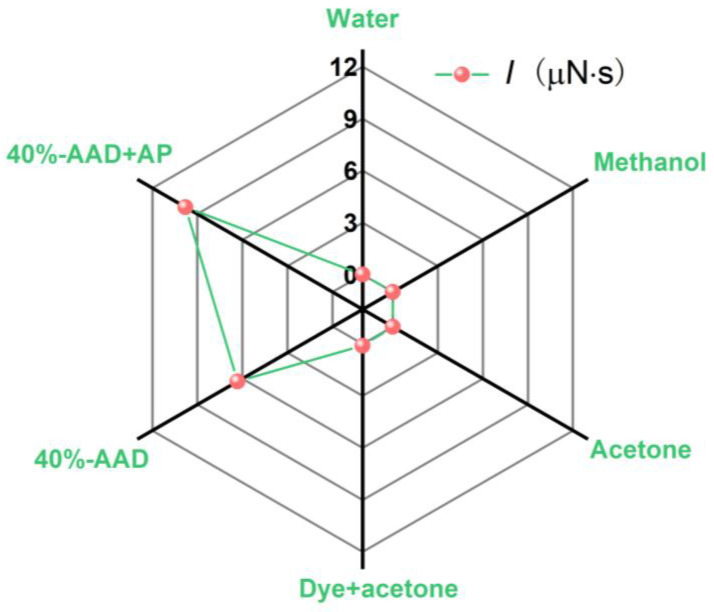
Experiment results of single-pulse impulse with different kinds of liquid propellants.

**Figure 9 micromachines-14-01219-f009:**
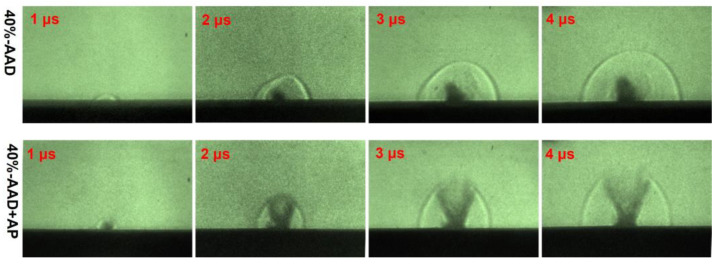
Schlieren images of laser ablation liquid propellant at different exposure times of 40%-AAD and 40%-AAD + AP propellant samples.

**Figure 10 micromachines-14-01219-f010:**
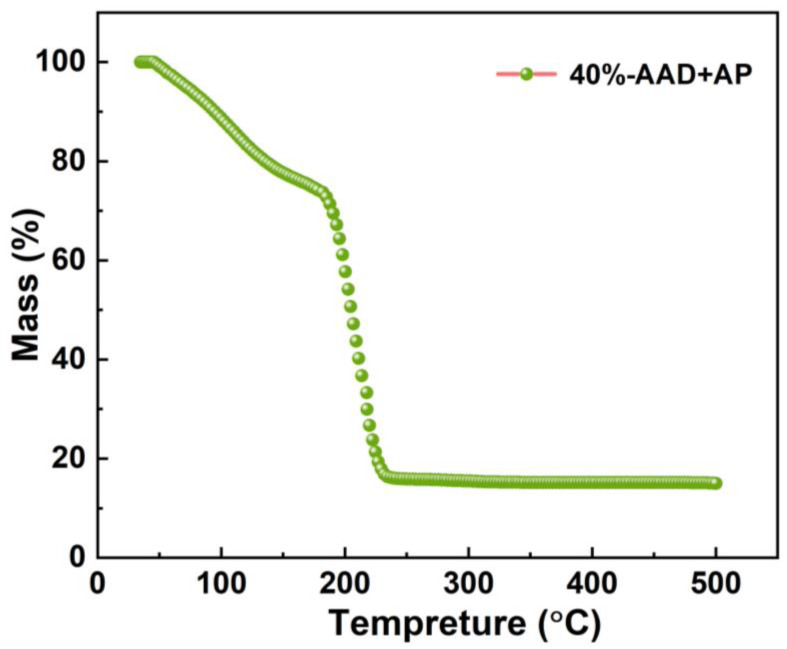
TG curves of 40%-AAD +AP Liquid propellant samples.

**Figure 11 micromachines-14-01219-f011:**
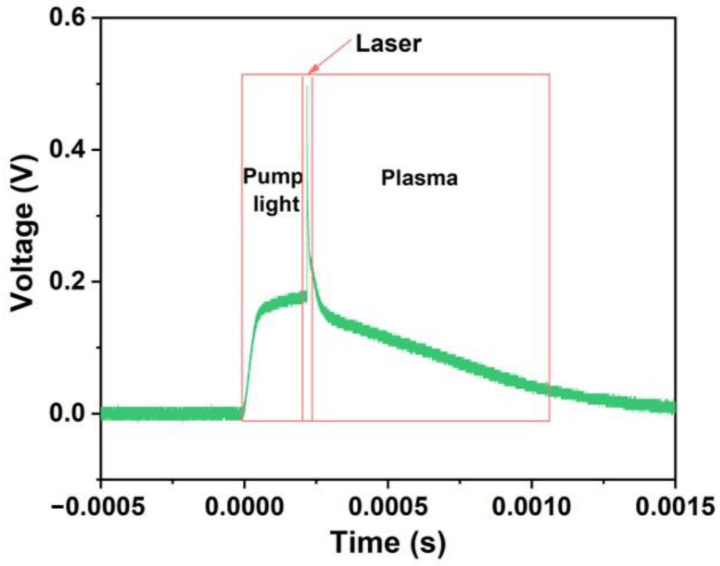
The pump light, laser and plasma signal of plume detected by photodiode.

**Table 1 micromachines-14-01219-t001:** Enthalpy variables of AAD solutions with different ratios.

Propellant Samples	Exothermic Peak Temperature (°C)	Enthalpy Variable (J/g)	Density (g/cm^3^)
0%-AAD	92.1	+108.78	0.84
20%-AAD	191.2, 194.5	−303.3	1.06
30%-AAD	194.5, 199.2	−313.24	1.08
40%-AAD	191.2	−345.7	1.10
50%-AAD	187.7	−368.3	1.12
60%-AAD	182.6	−371.7	1.27
80%-AAD	190.4	−425.44	1.31
100%-AAD	184.5	−432.18	1.37

**Table 2 micromachines-14-01219-t002:** Ablation performance toward liquid propellant in this work and in the literature.

Liquid Propellants	Combustion Chamber Volume (mm^3^)	Combustion Chamber Configuration	Ablation Performance	Application	References
ADN-based liquid propellant	π × 3.5^2^ × 5.3	Cavity	$\bar{F}$ = 0.2 N *I*_*sp*_ = 206 s	Chemical micropropulsion	[13]
ADN-based liquid propellant	3.5 × 4.5 × 0.3	Convergent-divergent nozzle	$\bar{F}$ = 30 mN	Chemical micropropulsion	[27]
ADN-based liquid propellant	20 × 20 × 0.05	Rectangular cavity	*I*_*sp*_ = 84.14 s *C*_*m*_ = 1070 dyne/W	Laser micropropulsion	[33]
ADN-based liquid propellant	12 × 12 × 5	Convergent-divergent nozzle	$\bar{F}$ = 1.3 N *I*_*sp*_ = 220 s	Chemical propulsion	[44]
1% carbon-doped glycerol	2.5 × 1.5 × 1.5	Cuboid	*C*_*m*_ = 154 dyne/W *I*_*sp*_ = 4 s	Laser micropropulsion	[45]
30–60% ADN-based liquid propellant	π × 52 × 25	Cavity	*I*_*sp*_ = 213–282 s	Chemical propulsion	[46]
ADN-based liquid propellant	1 × 5 × 25	Glass tube	$\bar{F}$ = 0.2–0.5 mN *I*_*sp*_ = 20–50 s	Chemical micropropulsion	[47]
ADN-based liquid propellant	π × 2.6252 × 7.25	Rectangular cavity	$\bar{F}$ = 1 N *I*_*sp*_ = 223 s	Chemical propulsion	[48]
Water	3.6 × 2 × 1	Convergent-divergent nozzle	$\bar{F}$ = 4.5 mN *I*_*sp*_ = 98 s *η =* 35.9%	Micropropulsion	[49]
ADN-based liquid propellant mixed with dye	0.2 × 0.2 × 0.2	Convergent-divergent nozzle	*I*_*sp*_ = 234.9 s *C*_*m*_ = 62.4 dyne/W *η =* 71.2%	Laser micropropulsion	In this work

## Data Availability

Not applicable.

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
