# Peer review of "Transmissive Mode Laser Micro-Ablation Performance of Ammonium Dinitramide-Based Liquid Propellant for Laser Micro-Thruster"

_micromachines, 2023, doi:10.3390/mi14061219_

Round 1
Reviewer 1 Report
Reviewer of manuscript MDPI Micromachines No. 2405507
“Transmissive Mode Laser Micro-Ablation Performance of Ammonium Dinitramide-Based Liquid Propellant for laser micro-thruster”
By Baosheng Du, Yongzan Zheng, Chentao Mao, Haichao Cui, Jianhui Han, Luyun Jiang, Jifei Ye and Yanji Hong
The manuscript covers an interesting and actual subject, namely space micro-propulsion systems for small satellites.
The English will need some corrections, but actions may occur after content modifications.
Abbreviations should be explained from the beginning (e.g. AAD?).
For a reader with not very deep knowledge it is not obvious how a laser ablation propulsion works. This should be explained early in the text, and compared to pure chemical propulsion, electrical propulsion (e.g. FEEP..). Schemes are welcome. Is the laser on ground or part of the thruster?
The experimental setup needs to be shown earlier in the paper. A thrust stand is mentioned in line 149, but the details come much further down the line.
Line 154: “ obtained by the calibration method..” Which method?
Line 255: as clearly shown in Fig. 4a… this is not clear at all
Figure 3 is very compressed, and the focal point may be shown in more detail.
Figure 4 should be better explained.
Line 329: AND-based??
Table 2: check and correct formatting
The English will need some corrections, but actions may occur after content modifications.
Author Response
Dear Reviewer,
Thank you for your letter and for the reviewer’s comments concerning our manuscript entitled “Transmissive Mode Laser Micro-Ablation Performance of Ammonium Dinitramide-Based Liquid Propellant for Laser Micro-thruster”. Those comments are all valuable and very helpful for revising and improving our paper. We have studied these comments carefully and have made correction according the comments, which we hope meet your approval. Revised portion are marked in red in the manuscript. The main corrections in the paper and the responds to the reviewer’s comments are as flowing:
Reviewer #1:
Comments No. 1: Abbreviations should be explained from the beginning (e.g. AAD?).
Response: Thanks to Reviewer for reminder, we have added the explanation “0.4 mL ADN solution dissolved in 0.6 mL of dye solution (40%-AAD)” in the manuscript.
Comments No. 2: For a reader with not very deep knowledge it is not obvious how a laser ablation propulsion works. This should be explained early in the text, and compared to pure chemical propulsion, electrical propulsion (e.g. FEEP..). Schemes are welcome. Is the laser on ground or part of the thruster?
Response: Thanks for your kind suggestion, we have explained the concept of laser ablation propulsion in lines 53-56 and summarized the shortcomings of chemical propulsion and electric propulsion systems in space propulsion missions in lines 45-48 of the manuscript. In addition, the laser is part of the micro-thruster in this work.
Comments No. 3: The experimental setup needs to be shown earlier in the paper. A thrust stand is mentioned in line 149, but the details come much further down the line.
Response: Thanks for your valuable suggestion, we have changed the position of experimental setup enable it to be shown earlier in the paper in Figure 1 in the manuscript. In addition, a torsional thrust stand composed by torsional arm, flexural pivots and electromagnetic damping is developed by our team for measuring ablation characteristics, which has been used in Ref.” Impacts of laser pulse width and target thickness on laser micro-propulsion performance, Plasma Sci. Technol. 24 (2022) 105504”.
Comments No. 4: Line 154: “obtained by the calibration method.” Which method?
Response: Your question is very good. Since we did not express it clearly. These system parameters can be obtained by the calibration method called free vibration method. The detailed process for the calibration can be found in Ref.” Impacts of laser pulse width and target thickness on laser micro-propulsion performance, Plasma Sci. Technol. 24 (2022) 105504”.
Comments No. 5: Line 255: as clearly shown in Fig. 4a… this is not clear at all.
Response: Thanks for your helpful reminder, we have revised “as clearly shown in Figure 4a” as “(Figures 4a-c)” in the manuscript.
Comments No. 6: Figure 3 is very compressed, and the focal point may be shown in more detail.
Response: Your question is very good. Since we did not show it clearly. We have marked the focus position in the Figure 1 in the manuscript.
Comments No. 7: Figure 4 should be better explained.
Response: Your question is very good. Since we did not express it clearly. We have further explained in detail the content of Figure 4 in lines 272-274 in the manuscript.
Comments No. 8: Line 329: AND-based?.
Response: We are very sorry for our incorrect writing, “AND-based” should be “ADN-based”, we have reviewed the full text carefully again, and revised all similar errors in the manuscript.
Comments No. 9: Table 2: check and correct formatting.
Response: We are very sorry for our incorrect formatting, we have reviewed the Table 2 carefully again, and revised all formatting errors in the manuscript.
Comments No. 10: The English will need some corrections, but actions may occur after content modifications.
Response: We are very sorry for our incorrect writing, we have reviewed the full text carefully again, and revised all errors in the manuscript.
Reviewer 2 Report
In this work, the transmissive mode laser micro-ablation performance of ammonium dinitramide (ADN)-based liquid propellants were researched in 200 µm scale combustion chamber for 1U laser micro-thruster design. The direct experimental evidence shows that outstanding laser energy deposition efficiency and rapidly heat release of ADN-based liquid propellants playing a crucial role in laser micro-ablation performance. What’s more, the underlying mechanisms of laser micro-ablation ADN-based liquid propellants was in-depth discussed. The results are very useful for improving the applications of pulsed laser micro-thruster.
Although there are some ambiguities in the text, it is recommended to accept this manuscript after a minor revision.
1. In the “Ablation characteristics measurement with torsion pendulum” section, what is “the energy coefficient (η) ” mean? Please give a brief explanation.
2. In the “Laser energy deposition characteristics of ADN-based liquid propellants” section, why choose acetone as the main component of liquid propellant? Please give a brief explanation.
3. The authors have summarized and compared the laser micro-ablation and propulsion performance of different liquid propellant in Table S2. But there is a lack of conclusion about the comparison.
4. What are the advantages of transmission mode laser micro-ablation liquid propellant for 1U laser-micro thruster?
5. The initial letter of “laser micro-thruster” in the title is not capitalized, please revise them.
6. There are some technical errors in text, like “AND” in line 329” can be revised as “ADN”, please revise them.
Author Response
Dear Reviewer,
Thank you for your letter and for the reviewer’s comments concerning our manuscript entitled “Transmissive Mode Laser Micro-Ablation Performance of Ammonium Dinitramide-Based Liquid Propellant for Laser Micro-thruster”. Those comments are all valuable and very helpful for revising and improving our paper. We have studied these comments carefully and have made correction according the comments, which we hope meet your approval. Revised portion are marked in red in the manuscript. The main corrections in the paper and the responds to the reviewer’s comments are as flowing:
Reviewer #2:
Comments No. 1: In the “Ablation characteristics measurement with torsion pendulum” section, what is “the energy coefficient (η) ” mean? Please give a brief explanation.
Response: Thanks to Reviewer for reminder, we have changed the energy coefficient in the text to the energy factor in the manuscript. The energy factor is an indicator that measures the kinetic energy generated by the ablation of liquid propellant by a unit of laser energy.
Comments No. 2: In the “Laser energy deposition characteristics of ADN-based liquid propellants” section, why choose acetone as the main component of liquid propellant? Please give a brief explanation.
Response: Your question is very good. Since we did not express it clearly. Acetone, as an important component of liquid propellant, has the following main functions: 1) It is miscible with other components and can dissolve solid ADN and oil-soluble near-infrared dyes; 2) As a part of propellant, its density is low, which is helpful for laser ablation of liquid propellant to produce high specific impulse; 3) Acetone is flammable and can be used as a fuel in liquid propellants.
Comments No. 3: The authors have summarized and compared the laser micro-ablation and propulsion performance of different liquid propellant in Table S2. But there is a lack of conclusion about the comparison.
Response: Thanks for your valuable suggestion, the conclusion “Although, their composites show better performance in ablation performance, our method present certain advantages in combustion chamber configuration and laser path design” has added in the manuscript in lines 422-424.
Comments No. 4: What are the advantages of transmission mode laser micro-ablation liquid propellant for 1U laser-micro thruster?
Response: Your question is very good. Since we did not express it clearly. The transmission mode laser micro-thruster has the following two advantages: 1) Simple and reliable optical path design; 2) Prevent splashing of liquid propellant from polluting the laser.
Comments No. 5: The initial letter of “laser micro-thruster” in the title is not capitalized, please revise them.
Response: We are very sorry for our incorrect writing, we have revised those errors in the manuscript.
Comments No. 6: There are some technical errors in text, like “AND” in line 329” can be revised as “ADN”, please revise them.
Response: We are very sorry for our incorrect writing, we have reviewed the full text carefully again, and revised all similar errors in the manuscript.